# Towards an All-Ireland Diamond Open Access Publishing Platform: The PublishOA.ie Project—2022–2024

Jane Mahony 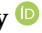

Trinity Long Room Arts & Humanities Research Institute, Trinity College Dublin, 2 Dublin, Ireland; mahonyjl@tcd.ie

**Abstract:** The Government of Ireland has set a target of achieving 100% open access to publicly funded scholarly publications by 2030. As a key element of achieving this objective, the PublishOA.ie project was established to evaluate the feasibility of establishing an all-island [Republic of Ireland and Northern Ireland] digital publishing platform for Diamond Open Access journals and monographs designed to advance best practice and meet the needs of authors, readers, publishers, and research funding organisations in Irish scholarly publishing. It should be noted in this context that there is substantial 'north–south' cooperation between public bodies in the Republic of Ireland and Northern Ireland in the United Kingdom, some of whom operate on what is commonly termed an 'all-island' basis. The project commenced in November 2022 and will run until November 2024, with the submission of a Final Report. This article originated as an interim project report presented in September 2023 at the PubMet2023 conference in Zadar, Croatia. The project is unique in its mandate to report on the feasibility of a shared platform that will encompass scholarly publishing across the two jurisdictions on the island of Ireland, which are now, post-Brexit, inside and outside the European Union (EU): the Republic of Ireland and Northern Ireland in the United Kingdom. The project is co-led by the Royal Irish Academy (RIA), Ireland's leading body of experts in the Sciences and Humanities, and the Trinity Long Room Hub Arts & Humanities Research Institute of Trinity College Dublin. There are sixteen partners and affiliates from universities and organisations from the island of Ireland. The feasibility study will be based on a review of the publishing practices in the island of Ireland, with gap analysis on standards, technology, processes, copyright practices, and funding models for Diamond OA, benchmarking against other national platforms, and specifications of the requirements, leading to the delivery of a pilot national publishing platform. A set of demonstrator journals and monographs will be published using the platform, which will be actively trialled by the partner publishers and authors. PublishOA.ie aims to deliver an evidence-based understanding of Irish scholarly publishing and of the requirements of publishers to transition in whole or in part to Diamond OA. This paper provides an interim report on progress on the project as of September 2023, ten months after its commencement.

**Keywords:** Ireland; open access books and journals; scholarly publishing; open research; Diamond Open Access



The Government of Ireland has set a target of achieving 100% open access to publicly funded scholarly publications by 2030 [1]. The *National Action Plan for Open Research 2022–2030* builds on a number of national policies and international recommendations, including the *National Principles on Open Access* (2012), the *European Commission Recommendation on access to and preservation of scientific information* (2018), the *National Framework on the Transition to an Open Research Environment* (2019), and the *UNESCO Recommendation on Open Science* (2021) [1].

Implementation of the *National Action Plan for Open Research* is led by Ireland's National Open Research Forum (NORF), established in 2017 and composed of expert representatives from policy, research funding, research performing, the library sector, research infrastructures, enterprise, and other key stakeholders in the research system across Ireland.

Implementation will be delivered under *Impact 2030: Ireland's Research and Innovation Strategy* [2] with the support of the Department of Further and Higher Education, Research, Innovation and Science.

As part of the overall action plan, the PublishOA.ie project has been tasked with conducting a feasibility study and pilot with a view to establishing a publicly owned, centralised national platform for Diamond Open Access (OA) journals and books, similar to the national platforms in Finland, Croatia, France, the Netherlands and Spain [1]. The platform will be designed to advance best practice and meet the needs of authors, readers, publishers, and funders in Irish scholarly publishing. The project is unique in its mandate to report on the feasibility of a shared platform that will encompass academic publishing across two jurisdictions, which are now, post-Brexit, inside and outside the EU: the Republic of Ireland and Northern Ireland in the United Kingdom. The project is co-led by the Royal Irish Academy (RIA) and Trinity Long Room Arts & Humanities Research Institute, working with a further sixteen institutions in the Republic of Ireland and Northern Ireland. It is funded by the National Open Research Forum (NORF). There are sixteen partners and affiliates from universities and organisations from the island of Ireland [1].

Diverse disciplinary perspectives, e.g., from the Irish Humanities Alliance, TCD Long Room Hub, Moore Institute at the University of Galway (Arts and Humanities), Irish Open Access Publishers (IOAPs) (multiple disciplines, including Social Sciences and STEM), Dublin Institute of Advanced Studies (DIASs) (Physics, Irish language), and the RIA (multiple disciplines, including STEM), will be integrated with international best practice. This will inform decisions on models and technical infrastructure tested and trialled to ensure their validity in the Irish context as well as their international validity.

Bibliodiversity and long-term sustainability are critical elements of the PublishOA.ie project. Publishing in Ireland must always regard the considerably larger and better-resourced publishing industry in the United Kingdom next-door, where Irish authors, writing in English, have historically been strongly represented [3]. The Republic of Ireland and Northern Ireland are both small countries, with current populations of approximately 5.28 million [4] and 1.9 million [5], respectively, totalling 7.18 million across the island.

Currently, there are 11 publishers of scholarly monographs in Ireland, but, by far, the majority of Irish interest monographs are published outside of Ireland, mainly in the UK, by large publishers such as Cambridge University Press (CUP), Oxford University Press, Liverpool University Press, and Manchester University Press. Similarly, several leading Irish interest journals are published outside of Ireland, including Irish Historical Studies, (CUP), the Irish Journal of Medical Science, (Springer), the Irish Journal of Economic and Social History (Sage), and Irish University Review (Edinburgh UP). The fact that so many Irish authors publish with UK university presses and commercial publishers clearly presents one of the key challenges of achieving 100% open access to publicly funded scholarly publications by 2030, since future adoption of open access publishing practices by these publishers is not yet clear [2].

In order to understand the current publishing landscape in Ireland, the first task of the PublishOA.ie project was to research, map, and publish the first digital directory of Irish publishers in the Republic of Ireland and Northern Ireland [6]. This was issued in June 2023. Together, the directory and map (Figures 1 and 2) list the activities and location of 183 publishers, publishing in both the English and Irish languages, and demonstrate the range of publishing, including fiction and non-fiction, scholarly, reference, professional, Irish language, children's and young people, and lifestyle and culture. As might be expected, there is a concentration of activities around the cities of Dublin and Belfast, but activity outside the metropolitan areas is also present.

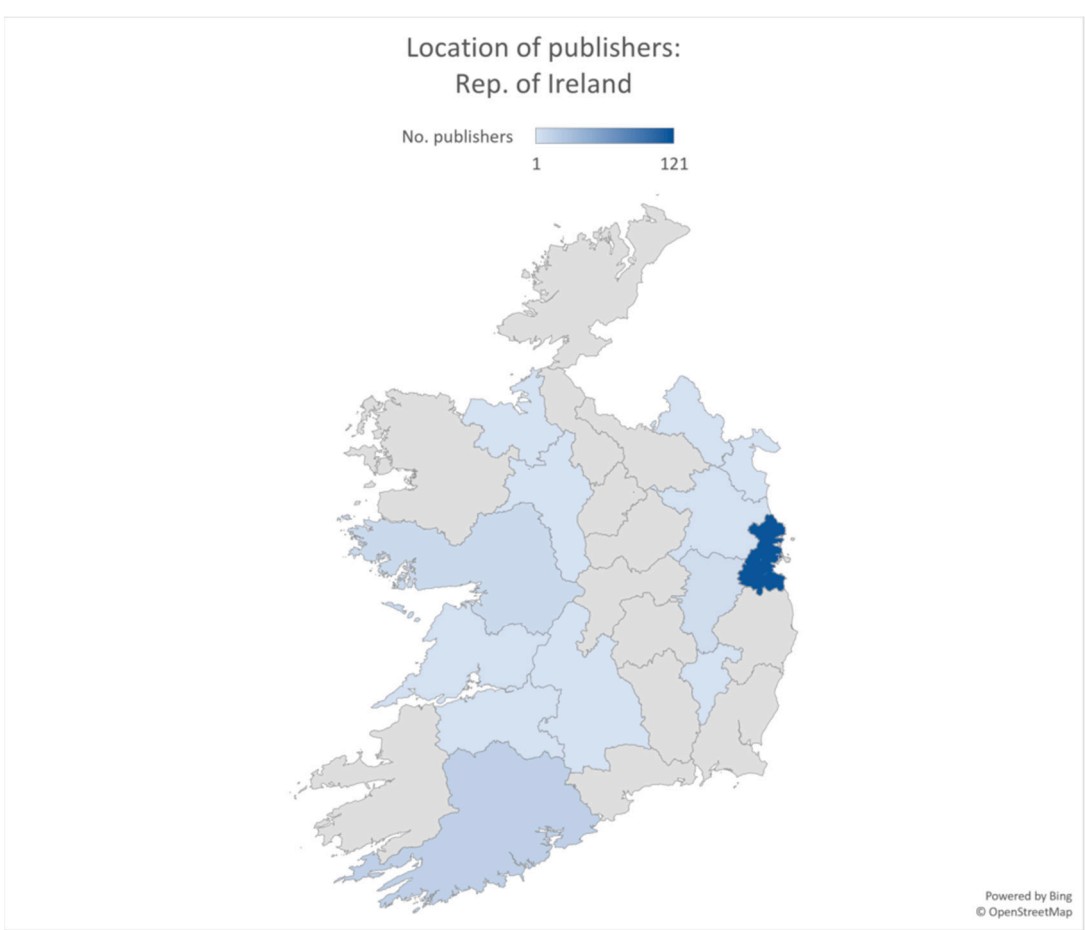

**Figure 1.** Map of publishing companies in the Republic of Ireland, June 2023.

PublishOA.ie is guided by the experience and expertise of members of our international advisory group. These include the Federation of Finnish Learned Societies (TSV.fi), Openjournals.nl, DOAJ, SDG Academy, and JSTOR, and other expert organisations (e.g., cOAlition S and DIAMAS). Consultation with these international organisations and institutions, all of which have demonstrable and significant experience in the field, is assisting us in analysing and discussing potential solutions to deploy, and where relevant, avail of existing developments. These discussions are proving especially valuable in the context of policy compliance (e.g., cOAlition S), technical advice (e.g., TSV and Openjournal.nl's use of Public Knowledge Project's Open Journal Systems), and potential OA content provision.

The intention is to build on and collaborate with the DIAMAS project [7], the European Union-funded consortium of 23 public service scholarly organisations from 12 European countries, whose objective is the delivery of an aligned, high-quality, and sustainable institutional OA scholarly publication ecosystem for the European Research Area. This includes setting a new shared and co-designed standard for Diamond OA publishing, a scholarly publication model in which journals and platforms are free for authors and readers (as defined by cOAlition S) [8].

PublishOA.ie is examining—with a view to adopting where assessed as relevant and appropriate—technical solutions and initiatives defined and designed by CoalitionS [9], Massachusetts Institute of Technology (MIT) Press [10], the EU's Open Science framework, defined as an approach to the scientific process that focuses on spreading knowledge as soon as it is available using digital and collaborative technology [11], and UNESCO's open access policy, defined as 'free access to scientific information and unrestricted use of electronic data for everyone. With open access, expensive prices and copyrights will no longer be obstacles to the dissemination of knowledge' [12].

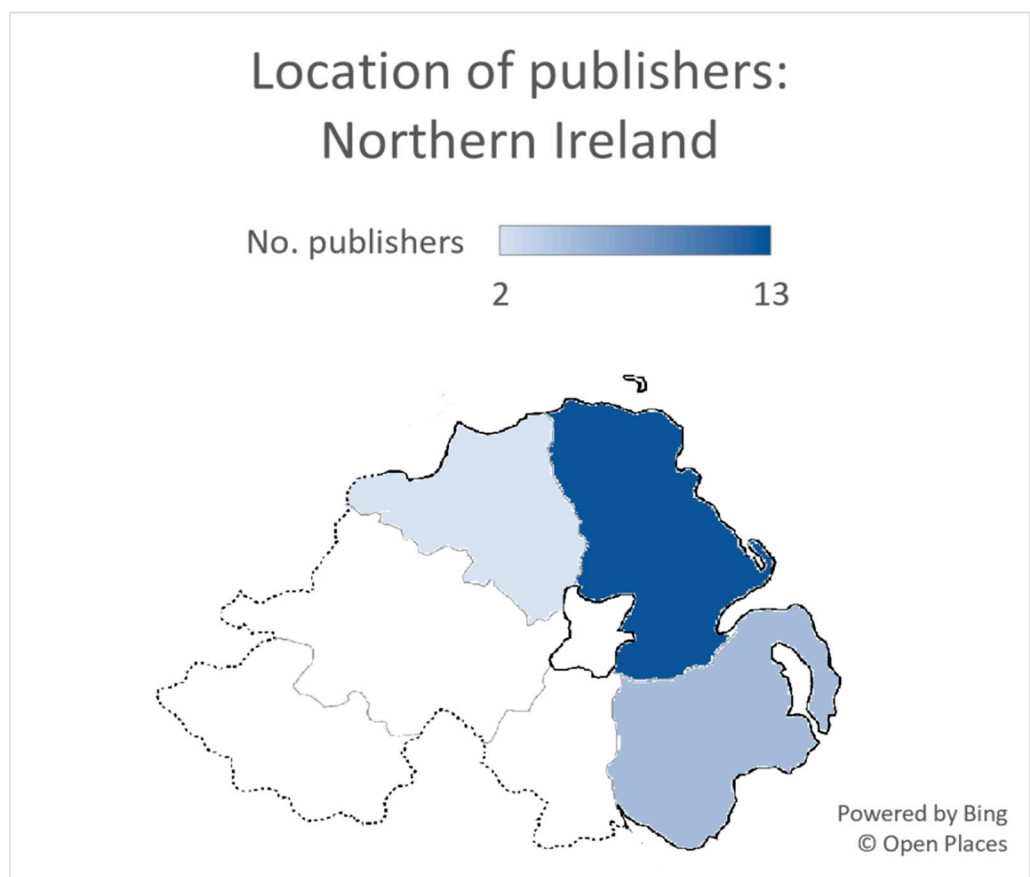

**Figure 2.** Map of publishing companies in Northern Ireland, June 2023.

As part of PublishOA.ie's mission to evaluate the feasibility of establishing a national publishing platform in Ireland, webinars were hosted in April, June, and July of 2023 for stakeholders, including publishers and librarians. In the first of the webinars, Open-Journals.nl and Journal.fi provided valuable technical and financial information on the operation of existing platforms in the Netherlands and Finland.

In July and August of 2023, platform providers Coko Foundation (Kotahi and Ketida), Janeway, Manifold, and the Public Knowledge Project's Open Journal Systems and Open Monograph Publishing (PKP OJS/OMP) provided demonstrations of their online publishing software and its application for editorial workflow and production. Information was shared on issues including platform maintenance and upgrading as well as data migration and journal onboarding to assist in building confidence. Other topics covered included training, IT and other support, funding sources and models, initial development budgets, annual running costs, subscription costs, as well as future planning, scaling, and sustainability.

The envisaged platform specification for Ireland will include the following elements:

- Articles will be open access on publication, accessible free of charge to anybody;
- Application of a clearly described peer review process;
- Author copyright retention;
- Application of creative commons licenses (CC BY 4.0 or similar);
- No Article Processing Charges for authors;
- Support for registration with the DOAJ/DOAB;
- Plan S compliance;
- Book publishing requirements compliance, including e-book formats, ONIX integration, author and reader engagement tools, social sharing, and tools to increase the visibility, accessibility, and impact of authors' work;
- Compliance with the 'Principles of Open Scholarly Infrastructure';

- Impact metrics and archiving—Directory of Open Access Books. Available online: https://doaj.org; https://www.doabooks.org/; https://openscholarlyinfrastructure.org/ (accessed on 16 March 2024).

As of September 2023, a set of national OA publishing principles for authors, developed in consultation with scholars, was being drafted, and it is expected in spring 2024. A set of OA publishing principles for publishers is also expected in spring 2024. A comparative technical specification, including the publishing platforms listed above, is also in preparation, complemented by an in-house examination of the technical parameters of seven further platforms: Digital Commons, Flax, Libero Reviewer, Meru, and PubPub.

PublishOA.ie aims to deliver an evidence-based understanding of Irish scholarly publishing and of the requirements of publishers to transition in whole or in part to Diamond OA. Further objectives include increasing awareness and engagement by policy makers of the system-level risks, challenges, and opportunities for a publicly owned centralised platform for the Diamond OA publication of journals and books. It is anticipated that the all-island approach will increase connections and relations within and between publishers, authors, librarians, scholars, and other stakeholders in scholarly publishing, thereby contributing to the Shared Island Initiative of the Government of Ireland [3]. This initiative aims to harness the full potential of the Good Friday Agreement of 1998 to enhance cooperation, connection, and mutual understanding on the island and engage with all communities and traditions to build consensus around a shared future. Key elements include working with the Northern Ireland Executive and the British Government to address strategic challenges faced on the island of Ireland; further developing the all-island economy, deepening north/south cooperation, and investing in the north west and border regions; and fostering constructive and inclusive dialogue and a comprehensive programme of research to support the building of a consensus around a shared future on the island.

At the time of the delivery of this paper at the PubMet Conference at the University of Zadar in September 2023, a number of key challenges had been identified and are being addressed as the project continues. These include the following:

- Scholarly publishers have articulated differing requirements for book and journal publishing. PublishOA.ie will, therefore, likely recommend and pilot two platforms (one for books, one for journals) to meet the objectives of the project. This was not anticipated at the outset of the project.
- Publisher engagement: publishers in Ireland are time- and resource-poor, especially in terms of staffing. Book publishers issue a mix of publications, and adoption of open access publishing is only one element of their activity.
- Future proofing has been identified as an issue of concern, encompassing the sustainability of the ongoing funding of both the Digital Repository of Ireland (DRI) and individual institutional repositories, and 'reference rot' (broken links), where the article or webpage resource identified by a URL no longer exists or has moved to another site. Reference rot also refers to the case whereby the resource identified has changed over time and evolved into a resource that bears no resemblance to the content originally referenced [13].
- The all-island mandate for a shared programme will require funding from the separate Governments of the Republic of Ireland and Northern Ireland. Since the UK's departure from the EU, so-called Brexit, the UK is no longer bound by EU law and directives (since this paper was delivered, the UK rejoined the Horizon Europe research and innovation programme in December 2023) [14].
- Bibliodiversity: support for smaller publishers will likely be required; the technical challenges of Diamond OA publishing may cause small publishers to outsource or divest themselves of publications to companies dedicated to this kind of publishing.
- Academic author buy-in: PublishOA.ie has identified that there are significant gaps in both knowledge and interest in open access publishing that may be mitigated by further and ongoing education and information.

## Conclusions

As stated in the *National Action Plan for Open Research*, 'Ireland supports the principle of full and immediate open access to research publications to ensure the widest possible dissemination of research. Making research publications openly and freely accessible contributed to an informed citizenship, the democratisation of knowledge, and maximises the impact of research processes and outputs' [1]. In line with Ireland's *National Framework on the Transition to an Open Research Environment*, there is a national objective that all Irish scholarly publications resulting from publicly funded research will be openly available by default [1]. The Diamond Open Access model—in which journals or publishers do not charge fees to either authors or publishers—has been identified as the favoured model.

As detailed in NORF's *National Open Research Landscape Report*, Ireland's progress in the area of open access research has been demonstrated by a steady growth in repository-mediated OA and an increase in publisher-mediated OA from 2015 onwards. Initiatives supporting the growth and uptake of open access have included a growing number of transformative agreements facilitated by IReL [1].

The PublishOA.ie project, which commenced in November 2022, will conclude in November 2024, when a Final Report will be issued. The project has been tasked with conducting a feasibility study and pilot with a view to establishing a publicly owned, centralised national platform for the Diamond OA publication of journals and books for Irish-based academic journals and publishers. The project working groups have identified a series of challenges and are currently engaged with the project's management and governance, technical feasibility study and pilot platform development, funding model analysis and reporting, archiving, and communications. While the project takes reference points from both our national and international partners and collaborators, it pays specific attention to the immanent conditions and contexts of the Irish publishing ecosystem. As noted earlier, there are significant gaps in both knowledge of and interest in open access publishing, and further education for stakeholders will, therefore, remain a key element of the continuing project. PublishOA.ie welcomes comments and suggestions from all interested parties.

**Funding:** The research for this article was funded by the PublishOA project of the National Open Research Forum of Ireland.

**Data Availability Statement:** Directory of Irish Publishers. Available online: https://PublishOA.ie.

**Conflicts of Interest:** The author declares no conflict of interest.

## Notes

1. The partner institutions are: Technological University of the Shannon (TUS); University of Galway (UG); Publishing Ireland; Dun Laoghaire Institute of Art, Design and Technology (IADT); Technological University Dublin (TU Dublin); Munster Technological University (MTU); University of Limerick (UL); South East Technological University (SETU). The affiliate institutions are the Irish Humanities Alliance (IHA); United Nations Sustainable Development Academy at University College Dublin (UN SDSN, UCD); Directory of Open Access Journals (DOAJ), Finnish Federation of Learned Societies; Openjournals.nl; Queen's University Belfast (QUB); the Dublin Institute of Advanced Studies (DIAS); JSTOR (Journal Storage).

2. See statement on Open Access from the UK Publishers Association. Available online: https://www.publishers.org.uk/our-work/open-access/. Note the statement that '[p]ublishers are committed to delivering on the Open Access agenda, but do require the necessary time, flexibility and funding to do so'. See also UK Research and Innovation (UKRI) policy statement on Open Access. Available online: https://www.ukri.org/publications/ukri-open-access-policy/uk-research-and-innovation-open-access-policy/ (accessed on 16 March 2024).

3. Available online: https://www.gov.ie/en/campaigns/c3417-shared-island/?referrer=https://www.gov.ie/sharedisland/ (accessed on 16 March 2024).

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
