# Peer review of "Towards an All-Ireland Diamond Open Access Publishing Platform: The PublishOA.ie Project—2022–2024"

_publications, doi:10.3390/publications12030019_

Round 1

Reviewer 1 Report

Comments and Suggestions for Authors

The discussion of the effects of Brexit on this project and the issues with establishing a national platform for two jurisdictions was very interesting and hopefully will continue in more detail in the Nov. 2024 report.

There were immediate questions about the necessity for publishing this progress update in a journal when it is so close to the delivery of the Nov. 2024 report. It would be helpful to understand the importance of publication if the abstract and/or introduction reflected in this paper given at the PubMet Conference at the University of Zadar in September 2023. As it stands, the only mention is in the middle of Page 5. 

Recommend not using terms like “international” when the focus of the feasibility study was relegated to Finland, Croatia, France, the Netherlands, and Spain. A reference for why the project only focuses on national programs in these listed countries when SciELO has successfully been around for over two decades and has some excellent documentation and reports on Creation, Operation, and Development of National Collections. 

https://scielo.org/en/about-scielo/programa-scielo-modelo-scielo-de-publicacao-e-rede-scielo/ 

It was good to see mention (p.5-6 bullets midway down) of that consideration of the challenges of staffing and resources to support the development, participation, and sustainability of a publishing infrastructure/platform will be addressed as the project continues. This is a common and major oversight in many “publicly owned” scholarly communication infrastructure projects.

Author Response

Please see attached file where I have incorporated the corrections suggested. Please also note addition of footnote for 'Shared Island Initiative' as suggested by Reviewer 2

Reviewer 2 Report

Comments and Suggestions for Authors

The paper presents an extended version of the communication delivered at the PUBMET2023 conference in Zadar (Croatia) last September. The description of the inspiring all-Ireland PublishOA.ie project is solid and well argued. The sole caveat is that a lot has happened of course in the Diamond OA context since this paper was delivered, and while the article provides a project snapshot with a Sep 2023 timestamp, it could make sense to add a note mentioning some of the most recent developments (Global Diamond OA Summit, unveiling of national-level Diamond OA strategies in various EU countries, consolidation of OA University Presses). This is for the author to decide, but given that a note was added to the text at a later stage (p. 6 "Since this paper was delivered, the UK rejoined the Horizon Europe research and innovation programme in December 2023") it could make sense to add a mention somewhere to the swiftly evolving Diamond OA landscape.

Other than that, most of the minor updates suggested below refer to the bibliography section, which is presently the weakest section of the paper (mainly from a formal perspective):

The PUBMET2023 origin of the communication is mentioned rather late in the paper itself (p. 5). This is ok since this piece is part of a special issue devoted to the event, but it could be useful to include a reference to the original communication in case readers wish to have a look at the slides delivered in Zadar, https://pubmet2023.unizd.hr/mahony-abstract/index.html

p. 5 "As of September 2023, a set of national OA publishing principles for authors, developed in consultation with scholars, was in draft and expected by the end of the year"
If these principles are already available, it could make sense to add a reference in the bibliography even if they were released after the 'timestamp' mentioned above

p. 5 "Shared Island Initiative of the Government of Ireland" might merit a reference? https://www.gov.ie/en/campaigns/c3417-shared-island/ (it's included in the original PUBMET communication)

Could it be worth adding a short funding acknowledgements section at the end of the paper? https://publishoa-ie.moodlecloud.com/course/view.php?id=8&section=4

References would need to be fine-tuned (and possibly cropped, it's unclear population stats should be part of the list; also, it's not good practice to post the URL for a webpage without any kind of standard reference info)

Author Response

Please see attached file where I have incorporated the corrections suggested - in bold type. Please also note addition of footnote for 'Shared Island Initiative' as suggested by Reviewer 2
